# Phenolic Composition of the Leaves of *Pyrola rotundifolia* L. and Their Antioxidant and Cytotoxic Activity

**DOI:** 10.3390/molecules25071749

**Published:** 2020-04-10

**Authors:** Katarzyna Szewczyk, Anna Bogucka-Kocka, Natalia Vorobets, Anna Grzywa-Celińska, Sebastian Granica

**Affiliations:** 1Department of Pharmaceutical Botany, Medical University of Lublin, 1 Chodźki Str., 20-093 Lublin, Poland; 2Department of Biology and Genetics, Medical University of Lublin, 4a Chodźki Str., 20-093 Lublin, Poland; anna.bogucka-kocka@umlub.pl; 3Department of Pharmacognosy and Botany, Faculty of Pharmacy, Danylo Halytsky Lviv National Medical University, 69 Pekarska Str., 79010 Lviv, Ukraine; vorobetsnatalia@gmail.com; 4Chair and Departament of Pneumonology, Oncology and Allergology, Medical University of Lublin, 20-093 Lublin, Poland; acelin@op.pl; 5Department of Pharmacognosy and Molecular Basis of Phytotherapy, Medical University of Warsaw, 02-091 Warsaw, Poland; sgranica@wum.edu.pl

**Keywords:** *Pyrola*, phenolics, antioxidants, cytotoxic activity, UHPLC–DAD–MS

## Abstract

The leaves of *Pyrola rotundifolia* L. were extracted in the mixed solvent of methanol/acetone/water (2:2:1, *v*/*v*/*v*) and investigated for their phytochemical analysis and biological activity. Total phenolic and flavonoid contents were determined spectrophotometrically. A high content of phenols (208.35 mg GAE/g of dry extract), flavonoids (38.90 mg QE/g of dry extract) and gallotannins (722.91 GAE/g of dry extract) was obtained. Ultra-high performance liquid chromatography diode array detector tandem mass spectrometry (UHPLC–DAD–MS) allowed for the detection of 23 major peaks at 254 nm. The extract was analyzed for its antioxidant capacity using 2,2-diphenyl-1-picryl-hydrazyl (DPPH^•^) and 2,2′-azinobis[3-ethylbenzthiazoline]-6-sulfonic acid (ABTS^•+^) radical scavenging, metal chelating power and β-carotene-linoleic acid bleaching assays. The examined extract showed moderate radical scavenging and chelating activity, and good inhibiting ability of linoleic acid oxidation (EC_50_ = 0.05 mg/mL) in comparison to standards. The cytotoxic effect in increasing concentration on five types of leukemic cell lines was also investigated using trypan blue vital staining. It was found that the analyzed extract induced the apoptosis of all the tested cell lines. Our findings suggest that the leaves of *P. rotundifolia* are a source of valuable compounds providing protection against oxidative damage, hence their use in traditional medicine is justified.

## 1. Introduction

The oxidative–antioxidant balance affects the proper functioning of homeostasis. Overproduction of reactive oxygen species (ROS) causes oxidative stress and is one of the most common reasons for homeostasis disorders. Oxidative stress, however, contributes mainly to civilization diseases, including neurological disorders [1], cancers [2], hypertension [3], atherosclerosis [4], diabetes [5], chronic obstructive pulmonary disease [6], and many others [7]. One of the ways of combating the undesirable effects of ROS is supplementation of exogenous antioxidants. A promising source of antioxidants are plants that have the innate ability to biosynthesize non-enzymatic antioxidants, which, in turn, have the ability to attenuate oxidative damage induced by reactive oxygen species [8].

The genus *Pyrola* L. (*Ericaceae*), comprising about 40 species [9], is mainly spread in the Circumboreal floristic region of the Holarctic. The species occur throughout Europe, western and central Asia, China and North America. In Europe, their range extends from the parallel of 68° N passing through northern Scandinavia, reaching as far as northern Spain and Italy, through Bulgaria, Crimea and the Caucasus, where the southernmost positions of the continent are located [10,11,12]. Many *Pyrola* L. species are used in traditional Eastern medicine due to their high nutritional value and valuable biological activities [13,14]. The aerial parts of *P. calliantha*, *P. decorata*, *P. incarnata*, *P. japonica*, *P. rotundifolia* and other species have been used to treat pulmonary hemorrhage, gastric hemorrhage, rheumatic arthritic diseases, kidney deficiency and urogenital diseases in traditional Chinese medicine [15,16,17,18,19,20]. *P. decorata*, as a tonifying agent, is a valuable element in many Chinese prescriptions for Alzheimer’s, Parkinson’s and other neurodegenerative diseases [21]. Moreover, *Pyrola* plants have confirmed pharmacological activities such as antibacterial [22,23], analgesic [18], anti-inflammatory [18,24], inhibitory effects of platelet aggregation [17], as well as vasodilating effects [25]. These properties are the result of many groups of active compounds in *Pyrola* L. The main groups of secondary metabolites that were found in the genus are flavonoids (quercetin, luteolin, rhamnetin and taxifolin and their glycosides) [12,13,17,24,26,27,28,29], phenols and their derivatives (arbutin, homoarbutin) [15,18,22,30], quinones (chimaphilin, 7′-hydroxychimaphilin, renifolin) [17,31,32], and triterpenoids (ursolic acid, oleanolic acid) [18,21,25,28,33].

*Pyrola rotundifolia* L., which is the subject of our research, is a perennial herbaceous plant [34] commonly known as round-leaved wintergreen [15]. Although the whole dried plant has been used for years in traditional medicine for the treatment of hypertension, rheumatic pain, tuberculosis, cancer and various inflammatory diseases [35], and is listed in Chinese Pharmacopoeia [36], the reports regarding the pharmacological activities of *P. rotundifolia* are limited [17,18,19,24,34,35,36].

Due to the importance of *Pyrola* species in traditional medicine and insufficient current knowledge about the antioxidant and cytotoxic activities among these plants, the purpose of the present study was to evaluate the biological properties of the leaves of *P. rotundifolia* and conduct their qualitative analysis using ultra-high performance liquid chromatography diode array detector tandem mass spectrometry (UHPLC-DAD-MS). Moreover, total content of polyphenols, flavonoids and gallotannins was determined. Radical scavenging of 2,2-diphenyl-1-picryl-hydrazyl (DPPH^•^) and 2,2′-azinobis[3-ethylbenzthiazoline]-6-sulfonic acid (ABTS^•+^), metal chelating power and β-carotene-linoleic acid bleaching assays were used to investigate antioxidant activity. The cytotoxicity of the 50% ethanol extract was estimated using trypan blue vital staining against five types of leukemic cell lines.

## 2. Results and Discussion

### 2.1. Phytochemical Analysis

In order to examine the potentially active compounds in *P. rotundifolia* with antioxidant potential, the total content of phenolic compounds, flavonoids and gallotannins was determined spectrophotometrically.

Phenolic compounds are one of the main classes of secondary metabolites that are responsible for many pharmacological activities. Their consumption can cause a decrease in the risk of diseases such as cancer and cardiovascular disfunction. Many researchers concluded that the supposed beneficial impact of polyphenols is often related to their antioxidant activity [37]. The phenolic content (TPC) was examined using Folin–Ciocalteu reagent and the results were expressed as gallic acid equivalents (GAE) per g of dry extract (DE) (Table 1). The TPC of leaves of *P. rotundifolia* was 208.4 ± 1.2 mg GAE/g DE. This value is comparable to those obtained by Zhang and co-authors [12] for *P. incarnata* from the Tahe region in northeast China (181.5 ± 3.7 mg GAE/g DW). The TPC values for samples of *P. incarnata* from other regions in northeast China ranged from 39.7 to 175.1 mg GAE/g DW.

Flavonoids, which constitute one of the largest groups of phenolic compounds, play various roles in the plants and in the human diet [38]. Many reports have suggested that they demonstrate biological properties, such as antibacterial, anti-inflammatory, antiallergic, hepatoprotective and antidiabetic [37,38]. However, the best described characteristic until now is their antioxidant potential [38]. The total flavonoid content of the leaves of *P. rotundifolia* was evaluated according to the previously described colorimetric method [39] and was expressed in quercetin equivalents (QE) per g of DE (Table 1). The observed flavonoid content was 38.9 ± 0.6 mg QE/g DE. The data for *P. rotundifolia* was higher than those obtained for *P. incarnata* from different regions of northeast China where the flavonoid contents varied from 2.5 to 22.1 mg of rutin equivalent per g of dry weight [12]. Total flavonoid content was also measured for *P. decorata*, *P. calliantha* and *P. renifolia* from different regions in China. Our results are most similar to those obtained for extracts from *P. decorata* collected in the Emei Mountains, Sichuan province (37.8 mg of rutin equivalent per g of dry weight) [13].

Tannins are one of the most widespread natural substances in nature. They are known as lipid peroxidation and lipooxygenases inhibitors, as well as a radical scavengers. Hydrolysable tannins, next to proanthocyanidins, are the major group of these compounds [40]. The total hydrolysable tannin content was measured using a very specific assay with rhodamine. It was found that *P. rotundifolia* leaves contain a high concentration of gallotannins with 722.9 μg GAE per gram of dry extract.

The next objective of our research was to perform qualitative analysis of the active compounds in the extract of the leaves of *P. rotundifolia*. The UHPLC–DAD–MS analysis of extracts allowed for the detection of 23 major peaks at 254 nm (Figure 1, Table 2). Compounds **2**, **3**, **5**, **6**–**8** and **14** displayed a single UV maximum at 274–278 nm, characteristic of a gallic acid moiety present in the structure. Compounds **2** and **3** showed a pseudomolecular ion in the MS spectrum at *m*/*z* = 331. The fragmentation of the base peak at *m*/*z* 331 showed signals of a gallic acid moiety at *m*/*z* = 169, and the neutral loss during the fragmentation was −162 amu, corresponding to a hexose unit. Thus, compounds **2** and **3** were characterized as galloylglucose isomers. Compound **5** showed a base peak signal in the MS spectrum at *m*/*z* = 483. The fragmentation of the major ion in the spectrum (483 amu) resulted in the production of a galloylglucose moiety at *m*/*z* = 331. Further fragmentation in the MS^3^ spectrum revealed the pattern-like compounds **2** and **3**. These observations led to the conclusion that **5** is a digalloylglucose isomer. Compounds **6** and **7** had a base peak ion in the MS spectrum at *m*/*z* = 437 and 325, respectively. Compound **6** was assigned as 6-*O*-galloylhomoarbutin based on previous reports on the phytochemical composition of *Pyrola incarnata* [30]. Compound **7** was identified as a galloylshikimic acid isomer based on characteristic fragmentation patterns as described by Abu-Reidah et al. [41]. Compound **1** had a pseudomolecular ion at *m*/*z* = 389 and UV–vis maximum at approximately 235 nm. It was assigned as monotropein, a compound that was previously described in several *Pyrola* species [26,42]. Compound **11**, with a major peak in the MS spectrum at *m*/*z* = 441 and fragmentation pattern showing peaks at *m*/*z* = 289 and 169, was identified as epicatechin gallate according to previous reports [30]. Compounds **10**, **12**, **13**, **15**, **16** and **22** had UV–vis spectra typical of flavonoids with maxima at approximately 260 and 350 nm. All of their fragmentation patterns revealed the presence of a strong signal at *m*/*z* = 301, suggesting that they contain quercetin as an aglycone. Based on comparisons made with available chemical standards [43], and according to the literature on the phytochemical composition of *Pyrola* species, compounds **10**, **12**, **13** and **16** were identified as quercetin 2′-*O*-galloylgalactoside, hyperoside, isoquercitrin and guajaverin, respectively [30]. Compound **15** was characterized as another quercetin, *O*-galloylhexoside, and compound **22** with a pseudomolecular ion at *m*/*z* = 585 was tentatively identified as quercetin *O*-galloylpentoside. The identity or partial identity of compounds **4**, **9**, **17**–**21** and **23** could not be resolved due to the unavailability of sufficient data. They were characterized by providing observed retention times, UV-vis maxima and MS data [11,28,30,41]. Although chimaphilin has been identified in several *Pyrola* species, including whole herb of *P. rotundifolia* [18], our study did not confirm the presence of this compound in the leaves of *P. rotundifolia*. This difference may be caused by a different habitat of plants or/and various methods of preparing extracts.

The quantitative evaluation of the molecules detected in the extract of the leaves of *P. rotundifolia* was performed based on the calibration curves of several compounds: epicatechin for the quantification of catechins, isoquercitrin for flavonoid derivatives, and gallic acid for galloyl derivatives. For unidentified compounds, we chose quercetin, because electrospray ionization (ESI) was similar to that of this standard. The areas of quantified compounds were determined at 275, 280 and 350 nm. The studied extract contained mostly quercetin (37.31 μg/mg DE) and gallic acid derivatives (18.65 μg/mg DE). Quercetin *O*-galloylhexoside (24.90 ± 1.17 µg/mg of DE) and galloylglucose isomer I (13.07 ± 0.11 µg/mg of DE) were the most abundant compounds. Similarly, Yao and co-authors [44] found mainly quercetin derivatives in *P. incarnata* extracts, although with a noticeably lower concentration in comparison to our investigation.

### 2.2. In Vitro Cytotoxicity Assay

Great attention has been recently given to natural compounds with established antioxidant effects and less toxicity in normal cells, because they appear promising for cancer prevention and treatment [45]. Plants and individual compounds isolated from them are one of the basic sources of potential chemopreventative and chemotherapeutic components. Because species belonging to the *Pyrola* L. are used in traditional medicine [13,14], in the next step of our study, the effect of the extract of *P. rotundifolia* leaves listed in the Chinese Pharmacopoeia [36] was investigated. The increasing concentrations of the ethanolic extract on human acute promyelocytic leukemia cell lines HL-60, HL-60/MX1 and HL-60/MX2, and acute lymphoblastic leukemia cell lines CEM/C1 and CCRF/CEM cancer cell lines were studied. The cytotoxicity was estimated using trypan blue vital staining. In the case of all leukemic cell lines exposed to the extract from the leaves of *P. rotundifolia*, a significant dose-dependent cytotoxic potential was observed (Figure 2).

The results for the cytotoxic effect, given in Table 3, showed that the tested extract significantly inhibited HL-60/MX1 and CCRF/CEM human leukemia cells with IC_50_ values of 3.3 and 6.2 μg/mL, respectively. Furthermore, the extract from the leaves showed a moderate cytotoxicity against three other lines—HL-60 with an IC_50_ = 12.3 ± 2.1 μg/mL, CEM/C1 with an IC_50_ = 16.9 ± 1.9 μg/mL and against HL-60/MX2 with an IC_50_ = 17.8 ± 2.4 μg/mL. The viability of the CCRF/CEM cell line after incubation with 1 μg/mL ethanolic extract of *P. rotundifolia* leaves were greatly affected (99% of nonviable cells) compared to untreated cells. Similar results were obtained at 300 μg/mL extract concentration for HL-60/MX1 (98.7% of nonviable cells). Control assays were performed using only the relevant volumes of medium and were considered as 100% of cell viability. Regardless of the examined extract concentration and type of cell line, the viability of the extract-treated cells differed significantly from that of the untreated cells (control) and was observed at 94.3 ± 0.04 to 1.0 ± 0.2% of the control viability.

Published papers on the cytotoxic activity of *Pyrola* species are limited. The cytotoxic effect of various extracts from the roots of *P. japonica* and the major isolated compound, chimaphilin, were tested against murine L1210 and human chronic myelogenous K562 leukemia cell lines [46]. Chimaphilin isolated from *P. incarnata* demonstrated inhibition of the viability of tested human breast cancer MCF-7 cells (IC_50_ = 43.30 μM) and caused mitochondrial dysfunction and caspase-dependent apoptosis [47].

Our study attempted to indicate a correlation between phenolic compound content and cytotoxicity against various leukemic cell lines. Pearson’s correlation coefficient was used to describe the mutual relationships (Table 4). A very strong positive correlation was found for total gallotannin content and HL-60, HL-60/MX1, HL-60/MX2 and CCRF/CEM cell lines (*r* = 0.9972, 0.9999, 0.9994 and 0.9999, respectively). A good negative correlation was observed for TPC and the above leukemic cell lines (*r* = −0.8053, −0.8386, −0.8278 and −0.8401, respectively). Our findings suggest that gallotannins may be responsible for the exertion the cytotoxicity activity.

Moreover, the presence of a large amount of quercetin derivatives may be responsible for the cytotoxic activity of *P. rotundifolia* leaves. It has been reported that these compounds may be considered as potential anticancer agents that have the ability to inhibit and prevent carcinogen-induced tumors [48].

### 2.3. Antioxidant Activity

It is known that reactive oxygen species (ROS), such as superoxide (O_2_^•-^), alkoxyl (RO^•^), peroxyl (ROO^•^), nitric oxide (NO^•^) and hydroxyl (HO^•^), apart from having some positive effects on the regulation of cell growth or synthesis of biologically important components, can also cause significant damage to cells. They can induce oxidation, which causes various degenerative diseases [37,38]. Although many synthetic antioxidants are currently available, it is important to search for new, natural compounds that could prevent oxidative stress. Antioxidation could be, among others, an effect of scavenging of free radicals, metal chelation and enzyme inhibition. Flavonoids, especially quercetin derivatives, which are most abundant in the leaves of *P. rotundifolia*, are powerful antioxidants with anti-inflammatory and anti-cancerogenic effects [49].

The measurement of antioxidant activity in our study was performed using four colorimetric protocols on a microplate scale in cell-free systems. The obtained results are presented in Table 5.

The examined extract exhibited moderate ability to scavenge the DPPH^•^ free radical in a concentration-dependent manner. For comparison, the radical scavenging activity of ascorbic acid was measured under the same conditions. The EC_50_ value for the extract of the leaves (0.2 ± 0.01 mg/mL) was almost three times higher than the one for ascorbic acid. Our result can be compared to the radical scavenging activity of *P. decorata*, *P. calliantha* and *P. renifolia* from various regions in China analyzed by Wang et al. [13]. Among these species, the highest ability was reported for *P. calliantha* (IC_50_ = 9.66 μg/mL), followed by *P. renifolia* (IC_50_ = 24.80 μg/mL) and *P. decorata* (IC_50_ = 37.11 μg/mL). Moreover, the extract from the aerial parts of *P. incarnata* possesses high antioxidant activity with an IC_50_ = 0.121 mg/mL [44]. The DPPH radical scavenging activity of *P. incarnata* from the different regions of China was also investigated by Zhang et al. [12]. They found that samples from eight sites from northeast China have moderate ability to scavenge DPPH^•^ (IC_50_ value from 0.106 to 0.282 mg/mL).

Smirnova and co-authors [50] examined the radical scavenging capacity of *P. rotundifolia*. Although the authors obtained a scavenging effect of 71% for the ethanolic extract, the value is difficult to compare because we do not know what part of the plant was used or what the concentration of the examined extract was.

The percentage of ABTS^•+^ radical reduction caused by the extract of *P. rotundifolia* was measured after 10 min of incubation at λ = 734 nm. In this assay, the reported value from the Trolox Equivalent Antioxidant Capacity (TEAC) protocol was 0.6 ± 0.03 mmol Trolox/g of dry extract, which is comparable to results obtained for *P. incarnata* from different regions in China (0.25–0.64 mmol Trolox/g) [12] and for *P. decorata* (696.78 ± 67.94 μmol Trolox/g) [13]. The extracts of *P. calliantha* and *P. renifolia* showed a weaker capacity to neutralize the ABTS^•+^ radical cation (984.64 and 818.96 μmol Trolox/g, respectively) [13].

In our study, we also showed that the extract of *P. rotundifolia* had a moderate ferrous ion chelating effect, and were thereby able to capture ferrous ions before ferrozine formation. As in case of the DPPH^•^ assay, the chelating activity of the extract was three times weaker than the activity of the standard used (EDTA, EC_50_ = 0.5 ± 0.1 mg/mL).

In the β-carotene bleaching protocol, the degree of linoleic acid oxidation is established by evaluating its oxidation products that attack β-carotene (bleaching of yellow color) [51]. The extract of *P. rotundifolia* leaves showed the inhibiting ability of linoleic acid oxidation (EC_50_ = 0.1 ± 0.03 mg/mL), comparable to butylated hydroxytoluene (BHT) which was used as positive standard (EC_50_ = 0.02 ± 0.01 mg/mL). Zhang and co-authors found that extracts of *P. incarnata* from different regions of China also inhibited the oxidation of linoleic acid with increasing concentrations (IC_50_ from 0.032 to 0.097 mg/mL) [12].

Many studies on the relationship between antioxidant capacity and total polyphenol content have been conducted. Some scientists found a strong correlation between these parameters [52], while others have not shown such correlation [53]. In our study, Pearson’s correlation coefficient was used to describe the correlation between the antioxidant capacity and the TPC, TFC and GTC content levels (Table 6). A very strong positive correlation was found between the IC_50_ values of DPPH^•^ and ABTS^•+^ radicals with total polyphenol content (*r* = 0.93 and 0.89, respectively), and also between DPPH^•^ radical and total flavonoid content (*r* = 0.98). In addition, GTC strongly negatively correlated with the ABTS^•+^ radical (*r* = −0.99). A good positive correlation was also observed between the IC_50_ values of β-carotene bleaching and total flavonoid content (*r* = 0.79). The obtained results showed that good antioxidant activity of the extract of *P. rotundifolia* leaves is not related only to total polyphenol content, but also to other compounds such as gallotannins and flavonoids.

## 3. Materials and Methods

### 3.1. Plant Material

The leaves of *Pyrola rotundifolia* L. were collected in the territory adjacent to the Yavoriv National Park (49°59′53.2″ N; 23°40′43.8″ E) (Ukraine). The identity of the species was confirmed by Natalia Vorobets. A voucher specimen was deposited in the Department of Pharmaceutical Botany (PR-150716).

The plant material was dried in air, shade and at an average temperature of 24.0 ± 0.5 °C for 6 days [54]. The loss of weight after drying was 6.53% ± 0.19 [54]. Fifty grams of dried, powdered and sieved through a 0.28-mm sieve leaves were extracted successively five times with methanol/acetone/water (2:2:1, *v*/*v*/*v*) (1 h each, solid-to-solvent ratio = 1:4) using an ultrasonic bath (Bandelin Sonorex Digitec DT 100H, Berlin, Germany). The extract was filtered, evaporated under reduced pressure, and then lyophilized using a freeze dryer apparatus (Free Zone 1 apparatus; Labconco, Kansas City, KS, USA) and 7.35 g of dried extract was obtained. For UHPLC analysis and antioxidant assays, 100 mg of obtained extract was dissolved in 10 mL 80% methanol and filtered through a 0.45 μm Chromafil syringe polyester membrane (Macherey-Nagel, Düren, Germany). For cytotoxic activity analysis, 100 mg of crude extract was dissolved in 10 mL 50% ethanol and filtered to give the stock solution (10 mg/mL).

### 3.2. Chemicals and Reagents

Water and formic acid for LC analysis were purchased from J.T. Baker (Deventer, Holland). HPLC grade acetonitrile, ascorbic acid, 2,2′-azino-bis-(3-ethyl-benzthiazoline-6-sulfonic acid) (ABTS^•+^), 2,2-diphenyl-1-picrylhydrazyl radical (DPPH^•^), 2,6-di-*tert*-butyl-4-methylphenol (BHT), ferrozine (3-(2-pyridyl)-5,6-bis(4-phenyl-sulfonic acid)-1,2,4-triazine), linoleic acid, rhodamine, Tween 40, and Folin–Ciocalteu reagent were from Sigma-Aldrich (St. Louis, MO, USA). Gallic acid and quercetin were obtained from Acros Organics (Geel, Belgium). Ethylenediaminetetraacetic acid (EDTA) and all other chemicals and solvents were of analytical grade and were purchased from Avantor-POCh (Gliwice, Poland).

### 3.3. Total Phenolic, Flavonoid, and Gallotannin Content

The total phenolic (TPC) and total flavonoid content (TFC) were determined using the colorimetric methods previously described [39]. TPC was examined as follows: 20 μL of the extract was added to 20 μL of Folin–Ciocalteu reagent (diluted with water 1:4 *v/v*). Then, 160 μL of sodium carbonate (75 g/L) was added, and the absorbance was measured after 20 min incubation at 750 nm using an ELISA Reader Infinite Pro 200F (Tecan Group Ltd., Männedorf, Switzerland). The TPC was expressed as mg of gallic acid equivalents per g of dry extract (mg GAE/g DE) using a calibration curve for gallic acid (0.001–0.008 mg/mL; *R*^2^ = 0.9908). For TFC determination, 20 μL of the extract was mixed with 60 μL of 96% ethanol, 4 μL of 1 M sodium acetate, 4 μL of 10% aqueous solution of aluminum chloride and filled with water to 200 μL. The absorbance was measured after 30 min incubation at 430 nm. The total flavonoid content was expressed as mg of quercetin equivalent per g of dry extract (mg QE/g DE) using a calibration curve for quercetin (0.05–0.20 mg/mL; *R*^2^ = 0.9999). Gallotannin content (GTC) was assayed on microplates according to the method described by Inoue and Hagerman [55] with some modifications. Five microliters of the examined extract was mixed with 6 μL of 0.4 N sulphuric acid followed by 24 μL of rhodamine. After 10 min, 8 μL of 0.5 N KOH was added, and after a following 3 min, 160 μL of water was added. The absorbance was measured after a 15 min incubation at 520 nm. The GTC was expressed as μg of gallic acid equivalent per g of dry extract (μg GAE/g DE) using a calibration curve for gallic acid (2–20 μg/mL; *R*^2^ = 0.9906).

### 3.4. UHPLC–DAD–MS Analysis

The UHPLC analysis was performed using an Ultimate 3000 series system (Dionex, Idstein, Germany) equipped with a dual low-pressure gradient pump with vacuum degasser, an autosampler, a column compartment, and a diode array detector coupled with Amazon SL ion trap mass spectrometer (Bruker Daltonik GmbH, Bremen, Germany). The separation of compounds in the analyzed extract was carried out with a Kinetex XB-C_18_ analytical column (100 mm × 2.1 mm × 1.9 µm), Phenomenex (Torrance, CA, USA). Column temperature was maintained at 25 °C. Elution was conducted using mobile phase A (0.1% HCOOH in deionized water) and mobile phase B (0.1% HCOOH in acetonitrile) with a multi-step gradient as follows: 0 min 5% B, 60 min 26% B, 70 min 50% B and finally 75 min 95% B. The flow rate was set to 0.3 mL/min during extract and standard analysis. Five microliters of each sample was introduced to the column by the autosampler. The column was equilibrated for 10 min between injections. UV–vis spectra were recorded in the range of 200–450 nm. Chromatograms were acquired at 254 nm. The eluate was introduced into the mass spectrometer without splitting. The ion trap Amazon SL mass spectrometer was equipped with an ESI interface. The parameters for the ESI source were set as follows: nebulizer pressure 40 psi; dry gas flow 9 L/min; dry temperature 300 °C; and capillary voltage 4.5 kV. Analysis was carried out using scan from *m*/*z* 70–2200. Compounds were analyzed in negative and positive ion mode. The MS^2^ and MS^3^ fragmentations were performed using Smart Frag mode. Compounds were tentatively identified based on the determination of their molecular mass, UV-vis spectra and fragmentation profiles in respect to the literature data on what compounds were previously detected or isolated from different *Pyrola* species. The search for potential matches was done using the Reaxys database.

#### Quantification of Major Flavonoids by UHPLC–DAD

The quantification of the compounds was performed using the analytical method reported in Section 3.4 and based on the calibration curves of several compounds: epicatechin for the quantification of catechins, isoquercitrin for flavonoid glucosides, gallic acid for galloyl derivatives, and quercetin for unidentified compounds. The areas of quantified compounds were determined at 275, 280 and 350 nm. For the quantification, accurately weighed 10 mg of DE was dissolved in 1 mL of methanol/water mixture (1:1, *v/v*). The solution was filtered through a 0.45 µm Polyvinyl Difluoride (PVDF) syringe filter and injected into the UHPLC (3 µL). Samples were analyzed in triplicate. Stock solutions of the standards were prepared by dissolving 1 mg of the compounds in 1 mL of methanol. The stock solutions were diluted in mobile phase to obtain solution at the concentration of 50 µg/mL. Different amounts of the standard were injected into the UHPLC column. The calibration curves were plotted based on the amount of injected compounds vs. peak area at 280 and 350 nm. The linear range was 50–400 ng/injection (R^2^ ≥ 0.99).

### 3.5. Cell Lines and Cell Culture

Human acute promyelocytic leukemia cell lines HL-60 (CCL-240**^™^**), HL-60/MX1 (CRL-2258**^™^**), and HL-60/MX2 (CRL-2257**^™^**), acute lymphoblastic leukemia cell lines CEM/C1 (CRL-2265**^™^)** and CCRF/CEM (CCL-119**^™^**) were used in the study. Cell lines were obtained from the American Type Culture Collection (ATCC^®^) 10801, University Boulevard Manassas, VA 20110, USA. Details regarding the cell lines used in the research have been described in our previous report [56]. The cells were kept in Roswell Park Memorial Institute (RPMI) 1640 Medium (Biomed, Lublin, Poland) with 10% fetal bovine serum (FBS) (PAA Laboratories) for HL-60/MX1, HL-60/MX2, CEM/C1, and CCRF/CEM, and 20% FBS for HL-60 cell lines, streptomycin and penicillin (100 U/mL PAA Laboratories), and 2.5 µg/mL amphotericin B (Gibco, Carlsbad, USA) at 37 °C in a humidified atmosphere of 5% CO_2_.

### 3.6. Analysis of Cell Viability

The cells used in all experiments were placed in 12-well plates (Sarstedt GesmbH, Wiener Neudorf, Austria) at an initial density of 1 × 10^6^ cells/mL. After incubation (24 h; 37 °C), the cell suspensions were incubated with extract of the leaves of *P. rotundifolia* at concentrations ranging from 1 to 1000 μg/mL. Next, 1 mL of cell suspension was centrifuged at 1000 rpm for 6 min and centrifuged cells were resuspended in 50 μL of PBS. Afterwards, 10 μL of cell suspension was mixed with 10 μL of 0.4% solution of trypan blue reagent (Bio-Rad, Hercules, CA, USA). The samples were incubated for 5 min. The cell viability was measured using a TC 10™ Automated Cell Counter (Bio-Rad). The experiment was done in triplicate. The IC_50_ (inhibitor concentration when cell viability is 50%) values were determined using MS Excel.

### 3.7. Antioxidant Activity Assays

All assays were made using 96-well microplates (Nunclon, Nunc, Roskilde, Denmark) and were measured in an ELISA Reader Infinite Pro 200F (Tecan Group Ltd., Männedorf, Switzerland).

The DPPH^•^ (2,2-diphenyl-1-picryl-hydrazyl) radical scavenging activity was measured by a previously described method [57]. DPPH solution (180 μL of freshly prepared 0.07 mg/mL solution) was mixed with 20 μL of the examined extract in various concentrations in microplates. The absorbance at 517 nm was monitored after 30 min incubation at 28 °C and the results were expressed as an EC_50_ value. Ascorbic acid was used as the control. The antiradical potential was also analyzed using the previously described ABTS^•+^ (2,2′-azinobis[3-ethylbenzthiazoline]-6-sulfonic acid) assay [57]. The absorbance was measured at 734 nm and the results were expressed as millimoles of Trolox equivalents per g of dry extract (TEAC).

The metal chelating activity was determined by the method described by Guo et al. [58] with some modifications. In this assay, 0.2 mM aqueous solution of ferric chloride and 0.5 mM aqueous solution of ferrozine were used. Twenty microliters of the 0.2 mM aqueous solution of ferric chloride (II) was mixed with 100 μL of extract at different concentrations. Next, 40 μL of 0.5 mM aqueous solution of ferrozine was added and microplates were shaken and incubated for 10 min in 24 °C. The absorbance was measured at 562 nm and the percentage of inhibition of ferrozine–Fe^2+^ complex formation was calculated using the following formula:% inhibition = (1 − (A_s_/A_c_)) × 100(1)
where A_c_ is the absorbance of the control (water instead of the extract), and A_s_ is the absorbance of the extract.

The results were presented as the concentration of the extract that causes metal chelating in 50% (EC_50_) calculated on the basis of the linear correlation between the inhibition of ferrozine–Fe^2+^ complex formation and the concentrations of the extract. EDTA was used as a positive control.

Antioxidant activity was also assayed using the β-carotene bleaching method described and modified by Deba and co-authors [51]. Twenty microliters of extract at different concentrations were mixed with freshly prepared β-carotene-linoleic acid emulsion, and incubated for 20 min at 40 °C. The absorbance was measured at 470 nm. BHT was used as a positive control.

### 3.8. Statistical Analysis

All results were expressed as means ± standard deviation (SD) of three independent experiments. One-way ANOVA with Tukey’s post hoc test was used for the statistical analysis of significance of differences between means. *p* values below 0.05 were accepted as statistically significant. The correlations were evaluated by calculating Pearson correlation coefficients. All the investigations were done using Statistica 10.0 (StatSoft Poland, Cracow, Poland).

## 4. Conclusions

More and more research in medicine is being devoted to reactive oxygen species (ROS). Currently, there is significant evidence that ROS induce oxidative damage in biomolecules, which is responsible for the formation of many diseases such as cancer [59]. Numerous studies have demonstrated that medicinal plants possess antioxidant potential from the redox features of phenolic compounds. Besides being good antioxidants, plants and their active constituents are good antitumor agents with cytotoxic activity against numerous cell lines [59].

In our study, we found that the leaves of *Pyrola rotundifolia* contain large amounts of biologically active compounds. Twenty-three compounds were detected in the UHPLC–DAD–MS analysis. Among them, isoquercitrin, guajaverin, monotropein, galloylglucose isomers, quercetin 2′′-*O*-galloylgalactoside and epicatechin gallate were described for the first time in the examined species.

From the obtained cytotoxic effect values of the ethanolic extract of *P. rotundifolia* against five leukemic cell lines, it can be concluded that the viability of cells considerably decreased with increasing doses of the examined extract. Many authors found that aqueous and ethanolic extracts have a major destructive effect on tumor cells as compared with nonpolar extracts [60]. It is known that phenolic compounds such as tannins and flavonoids are soluble in ethanol; therefore, they can be the major group of active components for the destruction of leukemic cells. Moreover, the examined extract had good antioxidant activity and can be considered as a radical inhibitor or scavenger. Although further details regarding the effect of *P. rotundifolia* leaves and their individual compounds against the leukemic cells examined in our study remain to be investigated, the present results suggest that the ethanolic extract would be beneficial in the chemoprevention of leukemia. Furthermore, as a rich source of phenolic compounds with good antioxidant properties, it can be used as a reliable source of natural antioxidants and may be used for food supplement production and in the pharmaceutical and cosmetic industry.

## Figures and Tables

**Figure 1 molecules-25-01749-f001:**
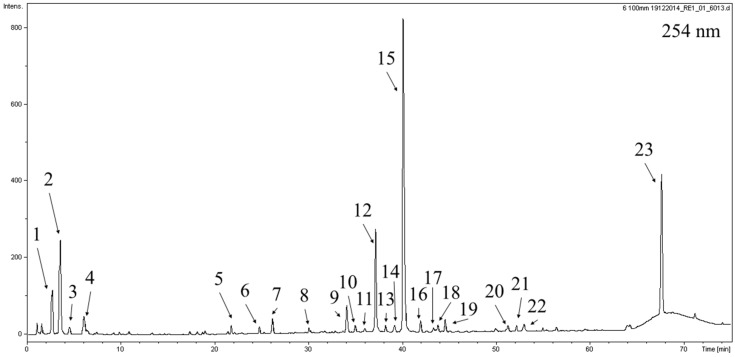
The ultra-high performance liquid chromatography diode array detector (UHPLC–DAD) chromatogram acquired at 254 nm with marked compounds. Peak numbers relate to those in Table 2.

**Figure 2 molecules-25-01749-f002:**
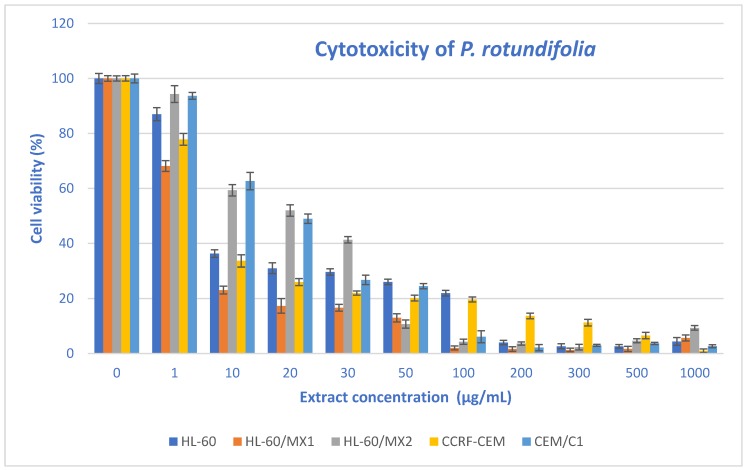
The valuation of the viability of various cell lines exposed to 24 h of increasing concentrations of an extract from *P. rotundifolia* leaves.

**Table 1 molecules-25-01749-t001:** The total phenolic (TPC), flavonoid (TFC) and gallotannin (GTC) content in the leaves of *P. rotundifolia*.

Phenolic Content	
**TPC (mg GAE/g DE)**	208.4 ± 1.2
**TFC (mg QE/g DE)**	38.9 ± 0.6
**GTC (μg GAE/g DE)**	722.9 ± 0.5

The results are expressed as mg/μg of gallic acid equivalent (GAE) and quercetin equivalent (QE) per g of dry extract (DE). Values are presented in mean ± standard error of the mean (SEM), *n* = 3.

**Table 2 molecules-25-01749-t002:** The ultra-high performance liquid chromatography diode array detector tandem mass spectrometry (UHPLC-DAD-MS^3^) data for compounds detected in the leaves of *P. rotundifolia*.

No.	Compound Name	Retention Time (min)	UV (nm)	[M − H]^−^ *m*/*z*	MS^2^ Ions	MS^3^ Ions	NL Detected (amu)	Compound Content (µg/mg of DE)
**1**	Monotropein ^t^	2.7	237	389	345, 227b, 209, 191, 179, 165, 147, 135	-	-	10.41 ± 0.07
**2**	Galloylglucose isomer I	3.5	278	331	314, 271, 211, 193, **169b**, 125	125b	162	13.07 ± 0.11
**3**	Galloylglucose isomer II	4.5	277	331	271, 207, 169b, 161, 125	-	-	0.69 ± 0.05
**4**	Unknown compound	6.1	282	331	285b, 241, 161, 123	-	-	0.91 ± 0.02
**5**	Digalloylglucose isomer	21.7	276	483	423, **331**, 313, 271b, 211, 169	271b, 241, 169	152	0.75 ± 0.03
**6**	6-*O*-Galloylhomoarbutin ^t^	24.7	274	437	313b, 271, 211, 169	-	-	0.87 ± 0.01
**7**	Galloylshikimic acid ^t^	26.1	277	325	205, 169b, 119, 101	-	-	1.24 ± 0.05
**8**	Gallic acid derivative	30.0	277	725	679b, 577, 517, 407, 331	-	-	0.61 ± 0.02
**9**	Unknown compound	34.0	224sh, 239, 302	349	259, 241, 229, 187b, 161	-	-	3.82 ± 0.12
**10**	Quercetin-2′′-*O*-galloylgalactoside ^s^	35.0	250, 261sh, 352	615	**463b**, 343, 301	343, 301b	152	0.67 ± 0.15
**11**	Epicatechin gallate ^t^	36.0	280	441	397, 331, **289b**, 271, 169	245b, 205	152	0.13 ± 0.01
**12**	Quercetin 3-*O*-galactoside (hyperoside) ^s^	37.1	254, 263sh, 351	463	343, **301b**, 179	271, 255, 179b, 151	162	9.35 ± 0.75
**13**	Quercetin 3-*O*-glucoside (isoquercitrin) ^s^	38.2	251, 261sh, 351	463	343, **301b**	-	162	0.72 ± 0.14
**14**	Gallic acid derivative	39.2	270	521	506, **359b**, 169	315, 169b	162	1.42 ± 0.07
**15**	Quercetin *O*-galloylhexoside	40.0	252, 263sh, 352	615	463, 343, 313, 301b	-	-	24.90 ± 1.17
**16**	Quercetin-3-*O*-arabinopyranoside (guajaverin) ^s^	41.9	251, 260sh, 352	433	343, **301b**	257, 179b, 151	132	0.89 ± 0.20
**17**	Unknown compound	43.2	-	391	259, 241, 229, 187b, 172	-	-	0.07 ± 0.01
**18**	Unknown compound	43.8	-	457	411b, 379, 337, 301, 217	-	-	0.25 ± 0.01
**19**	Unknown compound	44.5	226, 266	599	435, 313b, 285	-	-	0.61 ± 0.03
**20**	Unknown compound	51.2	242sh, 270	697	584, **535b**, 373, 355	373b, 355	162	0.57 ± 0.03
**21**	Unknown compound	52.2	-	963	777, 613b, 463, 299	-	-	0.49 ± 0.01
**22**	Quercetin-*O*-galloylpentoside	52.9	251, 263sh, 352	585	327, 285b, 255	505, 433, 301b, 283, 257, 229, 179, 151	-	0.78 ± 0.13
**23**	Unknown compound	67.6	224, 255, 269sh	533	515, 501, 472, 443, 384, 371b, 356, 335, 315	-	-	11.34 ± 0.05

DE—dry extract; s—comparisons with a chemical standard have been made; t—tentative assignment; b—base peak (the most abundant ion in recorded spectrum); in bold—ions subjected to MS^2^ or MS^3^ fragmentation; NL—normalization level (base peak intensity).

**Table 3 molecules-25-01749-t003:** The extrapolated IC_50_ values for the various leukemic cell lines.

Cell Line	IC_50_ (µg/mL)
HL-60	12.3 ± 2.1
HL-60/MX1	3.3 ± 0.4
HL-60/MX2	17.8 ± 2.4
CEM/C1	16.9 ± 1.9
CCRF/CEM	6.2 ± 0.7

**Table 4 molecules-25-01749-t004:** Pearson’s correlation coefficients between phenolic content and cytotoxicity of *P. rotundifolia* leaves.

*r* (p) for	HL-60 (IC_50_)	HL-60/MX1 (IC_50_)	HL-60/MX2 (IC_50_)	CCRF-CEM (IC_50_)	CEM/C1 (IC_50_)
**HL-60 (IC_50_)**	X	0.9983 (0.002)	0.9992 (0.001)	0.9981 (0.002)	0.6715 (0.329)
**HL-60/MX1 (IC_50_)**	0.9983 (0.002)	X	0.9998 (0.000)	0.9999 (0.000)	0.627 (0.373)
**HL-60/MX2 (IC_50_)**	0.9992 (0.001)	0.9998 (0.000)	X	0.9998 (0.000)	0.6421 (0.358)
**CCRF-CEM (IC_50_)**	0.9981 (0.002)	0.9999 (0.000)	0.9998 (0.000)	X	0.6249 (0.375)
**CEM/C1 (IC_50_)**	0.6715 (0.329)	0.627 (0.373)	0.6421 (0.358)	0.6249 (0.375)	X
**TPC (GAE)**	−0.8053 (0.195)	−0.8386 (0.161)	−0.8278 (0.172)	−0.8401 (0.160)	−0.1014 (0.899)
**TFC (QE)**	−0.4165 (0.583)	−0.4165 (0.583)	−0.3987 (0.601)	−0.4189 (0.581)	0.4471 (0.553)
**GTC (GAE)**	0.9972 (0.003)	0.9999 (0.000)	0.9994 (0.001)	0.9999 (0.000)	0.6146 (0.385)

*r*—correlation coefficients; p—probability; TPC—total phenolic content; TFC—total flavonoid content; GTC—total gallotannin content.

**Table 5 molecules-25-01749-t005:** The antioxidant activity (2,2-diphenyl-1-picryl-hydrazyl free radical scavenging assay (DPPH), 2,2′-azinobis[3-ethylbenzthiazoline]-6-sulfonic acid decolorization assay (ABTS), metal chelating (CHEL), β-carotene bleaching protocol) of extract from the leaves of *P. rotundifolia*. Each value is expressed as mean ± standard deviation (*n* = 3). Radical scavenging activity or DPPH for ascorbic acid (standard) EC_50_ = 0.08 ± 0.03 mg/mL; metal chelating activity (CHEL) for ethylenediaminetetraacetic acid (EDTA) (standard) EC_50_ = 0.45 ± 0.07 mg/mL; β-carotene bleaching assay for butylated hydroxytoluene (BHT) (standard) EC_50_ = 0.02 ± 0.01 mg/mL; DE—dry extract.

Antioxidant Activity	
DPPH (EC_50_ mg/mL)	0.2 ± 0.01
ABTS (TEAC mmol Trolox/g DE)	0.6 ± 0.03
CHEL (EC_50_ mg/mL)	1.4 ± 0.20
β-Carotene/linoleic acid (EC_50_ mg/mL)	0.1 ± 0.03

**Table 6 molecules-25-01749-t006:** Pearson’s correlation coefficients between phenolic content and antioxidant activity tests of *P. rotundifolia* leaves. Trolox Equivalent Antioxidant Capacity (TEAC), β-carotene/linoleic acid bleaching assay (β-C/LA).

*r* (p) for	DPPH (EC_50_)	ABTS (TEAC)	CHEL (EC_50_)	ß-C/LA (EC_50_)
DPPH (EC_50_)	X	0.6612 (0.339)	−0.6292 (0.371)	0.6595 (0.340)
ABTS (TE)	0.6612 (0.339)	X	0.1672 (0.833)	−0.1279 (0.872)
CHEL (EC_50_)	−0.6292 (0.371)	0.1672 (0.833)	X	−0.9992 (0.001)
ß-C/LA (EC_50_)	0.6595 (0.340)	−0,1279 (0.872)	−0.9992 (0.001)	X
TPC (GAE)	0.9300 (0.070)	0.8906 (0.109)	−0.2994 (0.701)	0.3371 (0.663)
TFC (QE)	0.9822 (0.018)	0.5087 (0.491)	−0.7638 (0.236)	0.7888 (0.211)
GTC (GAE)	−0.5925 (0.408)	−0.9961 (0.004)	−0.2534 (0.747)	0.2148 (0.785)

*r*—correlation coefficients; p—probability; TPC—total phenolic content; TFC—total flavonoid content; GTC—total gallotannin content.

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
