# Peer review of "Phenolic Composition of the Leaves of *Pyrola rotundifolia* L. and Their Antioxidant and Cytotoxic Activity"

_molecules, 2020, doi:10.3390/molecules25071749_

Round 1
Reviewer 1 Report
The manuscript entitled “Phenolics Composition of the Leaves of Pyrola rotundifolia L. and Their Antioxidant and Cytotoxic Activity”, even if not presenting a great impact, is enough interesting keeping a sufficient level of novelty and originality. All experimental parts have been performed with a good methodological approach.
I suggest few minor corrections:
- A general check of English language and typos
- Page 2, line 68 “the reports regarding pharmacological activities of P. rotundifolia are limited.” , add here references
- Substitute Figure 1 with a figure having a higher resolution.
Author Response
Dear Reviewer,
Thank you very much for Your recommendations. We have prepared a revised version of our manuscript, which has been modified in order to meet all your recommendations. Indeed, the comments were very useful to help us improving the manuscript. We hope that after the modifications we have made, the manuscript has been significantly improved.
To be in accordance with your recommendations, we have made the following changes:
The manuscript entitled “Phenolics Composition of the Leaves of Pyrola rotundifolia L. and Their Antioxidant and Cytotoxic Activity”, even if not presenting a great impact, is enough interesting keeping a sufficient level of novelty and originality. All experimental parts have been performed with a good methodological approach.
I suggest few minor corrections:
- A general check of English language and typos
English has been checked and corrected by native speaker (Kazimierz Kelles-Krauz).
- Page 2, line 68 “the reports regarding pharmacological activities of P. rotundifolia are limited.”, add here references
We have added references.
- Substitute Figure 1 with a figure having a higher resolution.
In our opinion, Figure 1 has a high resolution. The resolution depends on the form of the pdf file. We did the test by saving Figure 1 in two pdf at different resolutions and this confirms our opinion.
Sincerely Yours,
Authors

Reviewer 2 Report
Although the work shows, in general, some interesting aspects on biological activities of P. rotundifolia extracts, at the same time it presents some inaccuracies, particularly when the phytochemical composition is presented.
As indicated in the title the paper describes the phenolic composition of P. rotundifolia leaves and the evaluation of antioxidant and cytotoxic activities.
The Authors present evaluation of total phenolics, total flavonoids and total gallotannins with a simple spectrophotometric determinations, and then they give a preliminary qualitative (not even complete) evaluation of the phenolics in the extract by LC / MS.
A more complete and exhaustive evaluation of the phenolic fraction is necessary to improve the quality of the paper. A quantitation of compounds is required, as is usually done for this type of investigation.
Given the availability of standards and given the good separation of the compounds as shown in Figure 1, the authors should quantitate the various compounds of the extract using a calibration curves. This is the normal analytical procedure used for the presentation of such kind of data. This would make the paper much more meaningful and also justify the title.
Moreover, line 138: ‘Based on comparisons made with available chemical standards ........’ What kind of standards? From which source? Please clearly indicate.
Lines 142-144: 'The identity or partial identity of compounds ........ characterized by providing observed retention times, UV-Vis maxima and MS data'. What does it mean? Please specify the source of the data.
Lines 179-183: it is reported that P. japonica roots contain the naphthoquinone chimaphilin, and the same compound was extracted from P. incarnata. There are some evidences, some observations that this compound may also exist in P. rotundifolia?
Lines 185-187: it is reported that the volatile oil from P. herba show cytotoxic activity. Since this work is focussed on phenolic compounds, are there any particular phenols in the essential oil that justify its cytotoxic activity? And to justify this comparison?
Lines 179-187: it would be more appropriate that this part should be moved to Introduction.
In conclusion I found the paper incomplete in some parts, and requires a reorganization of the data and their presentation to reduce confusion and make the work more suitable for publication.
I think the paper requires major revisions and can be accepted for publication if all the suggestions will be performed.
Author Response
Dear Reviewer,
Thank you very much for Your recommendations. We have prepared a revised version of our manuscript, which has been modified in order to meet all your recommendations. Indeed, the comments were very useful to help us improving the manuscript. We hope that after the modifications we have made, the manuscript has been significantly improved.
To be in accordance with your recommendations, we have made the following changes:
Although the work shows, in general, some interesting aspects on biological activities of P. rotundifolia extracts, at the same time it presents some inaccuracies, particularly when the phytochemical composition is presented.
As indicated in the title the paper describes the phenolic composition of P. rotundifolia leaves and the evaluation of antioxidant and cytotoxic activities.
The Authors present evaluation of total phenolics, total flavonoids and total gallotannins with a simple spectrophotometric determinations, and then they give a preliminary qualitative (not even complete) evaluation of the phenolics in the extract by LC / MS.
A more complete and exhaustive evaluation of the phenolic fraction is necessary to improve the quality of the paper. A quantitation of compounds is required, as is usually done for this type of investigation.
Given the availability of standards and given the good separation of the compounds as shown in Figure 1, the authors should quantitate the various compounds of the extract using a calibration curves. This is the normal analytical procedure used for the presentation of such kind of data. This would make the paper much more meaningful and also justify the title.
The quantification of major flavonoids (10, 12, 13, 15, 16 and 22) detected in the investigated plant material was included in the revised version of the manuscript. Isoquercitrin was used as a standard for all compounds as they were quercetin derivatives. Although, other chemical standards were available for the comparison of retention times and UV-Vis and MS data the quantification was not performed for all compounds identified due to not enough amounts of standards for development of quantification procedure.
Moreover, line 138: ‘Based on comparisons made with available chemical standards ........’ What kind of standards? From which source? Please clearly indicate.
Citation for the paper were standards were isolated and identified was added (Granica, S., Hinc, K., Flavonoids in aerial parts of Persicaria mitis (Schrank) Holub , Biochemical Systematics and Ecology 61 (2015) 372-375)
Lines 142-144: 'The identity or partial identity of compounds ........ characterized by providing observed retention times, UV-Vis maxima and MS data'. What does it mean? Please specify the source of the data.
Citations for proper literature was added and the following statement was added in the material and methods section ‘Compounds were tentatively identified based on the determination of their molecular mass, UV-Vis spectra and fragmentation profiles in respect to the literature data on what compounds were previously detected or isolated from different Pyrola species. The search for potential matches was done using Reaxys database.’
Lines 179-183: it is reported that P. japonica roots contain the naphthoquinone chimaphilin, and the same compound was extracted from P. incarnata. There are some evidences, some observations that this compound may also exist in P. rotundifolia?
We reevaluated the obtained data by UHPLC-DAD-MS analysis. We were looking for chimaphilin but no peak that could be assigned to that compound was detected.
Lines 185-187: it is reported that the volatile oil from P. herba show cytotoxic activity. Since this work is focussed on phenolic compounds, are there any particular phenols in the essential oil that justify its cytotoxic activity? And to justify this comparison?
It is known that essential oil is a complex combination of a variety of chemical compounds, any of which may play a role in inhibition cell growth. As the Authors of cited publication have written, it is not clear as to which of the compounds contribute most significantly to antitumor activity. Therefore, compounds of volatile oil from Pyrola should be further investigated in order to elucidate the individual antitumor activity of each compound. Moreover, the Authors identified only 12 compounds in the Pyrola oil, mainly sesquiterpene alcohols.
Lines 179-187: it would be more appropriate that this part should be moved to Introduction.
Some of the information is in the Introduction, but in our opinion these lines are appropriate for Discussion section.
Sincerely Yours,
Authors

Reviewer 3 Report
In this paper, the authors presented interesting findings that could be repurposed in other fields for identifying bioactive compounds for suitable drug development. I think the paper is acceptable, however, there are few things that the authors could correct to make their paper suitable. 1. in line 403, the sentence with... it can be used as a redibly source of natural antioxidants. The word redibly is wrong, it should be reliable or credible.
Fig 2. can be improved by adding the scale to both the x and y-axis.
Author Response
Dear Reviewer,
Thank you very much for Your recommendations. We have prepared a revised version of our manuscript, which has been modified in order to meet all your recommendations. Indeed, the comments were very useful to help us improving the manuscript. We hope that after the modifications we have made, the manuscript has been significantly improved.
To be in accordance with your recommendations, we have made the following changes:
In this paper, the authors presented interesting findings that could be repurposed in other fields for identifying bioactive compounds for suitable drug development. I think the paper is acceptable, however, there are few things that the authors could correct to make their paper suitable. 1. in line 403, the sentence with... it can be used as a redibly source of natural antioxidants. The word redibly is wrong, it should be reliable or credible.
We have corrected it.
Fig 2. can be improved by adding the scale to both the x and y-axis.
We have changed Figure 2.
Sincerely Yours,
Authors

Round 2
Reviewer 2 Report
The Authors did not completely address some questions:
Given the availability of standards and given the good separation of the compounds as shown in Figure 1, the authors should quantitate the various compounds of the extract using a calibration curves. This is the normal analytical procedure used for the presentation of such kind of data. This would make the paper much more meaningful and also justify the title.
The quantification of major flavonoids (10, 12, 13, 15, 16 and 22) detected in the investigated plant material was included in the revised version of the manuscript. Isoquercitrin was used as a standard for all compounds as they were quercetin derivatives. Although, other chemical standards were available for the comparison of retention times and UV-Vis and MS data the quantification was not performed for all compounds identified due to not enough amounts of standards for development of quantification procedure.
The authors only quantitated some of the compounds detected in the extract and precisely the derivatives of quercetin, not considering the other abundant compounds.
The used approach is not complete as it is the incorrect way to present such kind of data! The authors should consider literature paper on this topic and refers to the analytical procedure.
It is not necessary to identify the other unknown compounds with the available standards, but simply quantitate them by using specific commercially available standards as was done for quercetin derivatives.
So for example, compounds 1, 2,… that are derivatives of gallic acid should be quantified with a standard of gallic acid, galloyl glucose, catechin gallate… arbutin.
Unknown compounds could be quantitate by using quercetin as a standard.
All these kind of information must be then inserted in the paper to perform an exhaustive quantitation of compounds.
Lines 179-183: it is reported that P. japonica roots contain the naphthoquinone chimaphilin, and the same compound was extracted from P. incarnata. There are some evidences, some observations that this compound may also exist in P. rotundifolia?
We reevaluated the obtained data by UHPLC-DAD-MS analysis. We were looking for chimaphilin but no peak that could be assigned to that compound was detected.
Please insert this observation in the text.
Lines 185-187: it is reported that the volatile oil from P. herba show cytotoxic activity. Since this work is focussed on phenolic compounds, are there any particular phenols in the essential oil that justify its cytotoxic activity? And to justify this comparison?
It is known that essential oil is a complex combination of a variety of chemical compounds, any of which may play a role in inhibition cell growth. As the Authors of cited publication have written, it is not clear as to which of the compounds contribute most significantly to antitumor activity. Therefore, compounds of volatile oil from Pyrola should be further investigated in order to elucidate the individual antitumor activity of each compound. Moreover, the Authors identified only 12 compounds in the Pyrola oil, mainly sesquiterpene alcohols.
Since the work is focused on the phenolic compounds of P rotundifolia and since the essential oil of the different species P. herba does not contain phenolic compounds, the sentence reported at lines 185-187 makes absolutely no sense.
It would be better to move this sentence to introduction as a general information on Pyrola spp.
In conclusion the Authors do not address some important questions and I consider the paper not suitable for publication.
Author Response
Dear Reviewer,
Thank you very much for Your comments. We have prepared a revised version of our manuscript, which has been modified in order to meet all your recommendations. Indeed, the comments were very useful to help us improving the manuscript. We hope that after the modifications we have made, the manuscript has been significantly improved.
To be in accordance with your recommendations, we have made the following changes:
The Authors did not completely address some questions:
The authors only quantitated some of the compounds detected in the extract and precisely the derivatives of quercetin, not considering the other abundant compounds.
The used approach is not complete as it is the incorrect way to present such kind of data! The authors should consider literature paper on this topic and refers to the analytical procedure.
It is not necessary to identify the other unknown compounds with the available standards, but simply quantitate them by using specific commercially available standards as was done for quercetin derivatives.
So for example, compounds 1, 2,… that are derivatives of gallic acid should be quantified with a standard of gallic acid, galloyl glucose, catechin gallate… arbutin.
Unknown compounds could be quantitate by using quercetin as a standard.
All these kind of information must be then inserted in the paper to perform an exhaustive quantitation of compounds.
We agree with Your suggestions. Unfortunately, during this challenging time, we have limited access to the laboratory, so it was diffcult for us to do additional analyses. What is more, the LC-MS analysis was performed in one city and some standards were available in another. Anyway, now we have managed to do this, therefore in the revised version of manuscript, we included new results of quantitative analysis.
Lines 179-183: it is reported that P. japonica roots contain the naphthoquinone chimaphilin, and the same compound was extracted from P. incarnata. There are some evidences, some observations that this compound may also exist in P. rotundifolia?
Please insert this observation in the text.
We have added this in the text.
Lines 185-187: it is reported that the volatile oil from P. herba show cytotoxic activity. Since this work is focussed on phenolic compounds, are there any particular phenols in the essential oil that justify its cytotoxic activity? And to justify this comparison?
Since the work is focused on the phenolic compounds of P rotundifolia and since the essential oil of the different species P. herba does not contain phenolic compounds, the sentence reported at lines 185-187 makes absolutely no sense.
We have delated these sentences.
In conclusion the Authors do not address some important questions and I consider the paper not suitable for publication.
Sincerely Yours,
Authors
